# Prognostic Value of Metabolic Imaging Data of ^11^C-choline PET/CT in Patients Undergoing Hepatectomy for Hepatocellular Carcinoma

**DOI:** 10.3390/cancers13030472

**Published:** 2021-01-26

**Authors:** Matteo Donadon, Egesta Lopci, Jacopo Galvanin, Simone Giudici, Daniele Del Fabbro, Ezio Lanza, Vittorio Pedicini, Arturo Chiti, Guido Torzilli

**Affiliations:** 1Department of Biomedical Science, Humanitas University, Via Rita Levi Montalcini 4, 20090 Pieve Emanuele, Milan, Italy; arturo.chiti@hunimed.eu; 2Department of Hepatobiliary and General Surgery, IRCCS Humanitas Research Hospital, Via Manzoni 56, 20089 Rozzano, Milan, Italy; jacopo.galvanin@humanitas.it (J.G.); simone.giudici@humanitas.it (S.G.); daniele.del_fabbro@humanitas.it (D.D.F.); 3Department of Nuclear Medicine, IRCCS Humanitas Research Hospital, Via Manzoni 56, 20089 Rozzano, Milan, Italy; egesta.lopci@humanitas.it; 4Department of Radiology, IRCCS Humanitas Research Hospital, Via Manzoni 56, 20089 Rozzano, Milan, Italy; ezio.lanza@humanitas.it (E.L.); vittorio.pedicini@humanitas.it (V.P.)

**Keywords:** hepatocellular carcinoma, hepatectomy, ^11^C-choline PET/CT, metabolic imaging, metabolic tumor volume

## Abstract

**Simple Summary:**

Few data are available for the use of ^11^C-choline positron emission tomography/computed tomography (PET/CT) in patients with hepatocellular carcinoma (HCC). The aim of the study was to analyze the clinical impact of this metabolic imaging in patients with HCC candidates for hepatectomy. Seven parameters were recorded for PET/CT in 60 patients. The Cox regression for overall survival (OS) showed that Barcelona stages (*p* = 0.003) and metabolic tumor volume (MTV) (*p* = 0.026) were the only factors independently associated with OS and furthermore, curve analysis revealed MTV ability in predicting OS. Patients with MTV ≥ 380 had worse OS (*p* = 0.015). The use of ^11^C-choline PET/CT allows for better prognostic refinement in patients undergoing hepatectomy for HCC: integration of such metabolic modality into HCC staging system should be considered.

**Abstract:**

^11^C-choline positron emission tomography/computed tomography (PET/CT) has been used for patients with some types of tumors, but few data are available for hepatocellular carcinoma (HCC). We queried our prospective database for patients with HCC staged with ^11^C-choline PET/CT to assess the clinical impact of this imaging modality. Seven parameters were recorded: maximum standardized uptake value (SUVmax), mean standardized uptake value (SUVmean), liver standardized uptake value (SUVliver), metabolic tumor volume (MTV), photopenic area, metabolic tumor burden (MTB = MTVxSUVmean), and SUVratio (SUVmax/SUVliver). Analysis was performed to identify parameters that could be predictors of overall survival (OS). Sixty patients were analyzed: fourteen (23%) were in stage 0-A, 37 (62%) in stage B, and 9 (15%) in stage C of the Barcelona classification. The Cox regression for OS showed that Barcelona stages (HR = 2.94; 95%CI = 1.41–4.51; *p* = 0.003) and MTV (HR = 2.11; 95%CI = 1.51–3.45; *p* = 0.026) were the only factors independently associated with OS. Receiver operating characteristics curve analysis revealed MTV ability in discriminating survival (area under the curve (AUC) = 0.77; 95%CI = 0.57–097; *p* < 0.001: patients with MTV ≥ 380 had worse OS (*p* = 0.015)). The use of ^11^C-choline PET/CT allows for better prognostic refinement in patients undergoing hepatectomy for HCC. Incorporation of such modality into HCC staging system should be considered.

## 1. Introduction

Hepatocellular carcinoma (HCC) is the fifth most common cancer and the third cause of cancer-related deaths worldwide [1]. Its treatment remains a global health issue because of its incidence and because of the complexity of its management. Apart from liver transplantation, which is considered the standard of care for patients who satisfy specific inclusion criteria [2], most patients receive loco-regional treatments such as hepatectomy, thermo-ablation, and trans-arterial therapies. However, these loco-regional treatments are burdened of disease recurrence up to 70% at five years [3,4,5,6]. This indicates the need of developing new clinical strategies to refine the workup of patients awaiting loco-regional treatments. Positron emission tomography/computed tomography (PET/CT) represents a molecular imaging modality applied for different types of cancer [7]. In general, the tracer mostly used before and after cancer treatments is 18F-fluorodeoxyglucose (FDG), which has demonstrated a suboptimal diagnostic accuracy for HCC but a potential role in disease prognostication [7,8,9,10]. Consequently, some authors reported new promising experiences with other radiopharmaceuticals, including radiolabelled choline [11,12,13,14,15], whose biological background relies on the increased activity of the key enzyme choline kinase in tumor cells compared with normal liver parenchyma, which is responsible for the entrapment of choline in HCC [16,17]. We previously reported the role of ^11^C-choline PET/CT in the pre-therapeutic work up of patients with HCC [18]. In this study, we sought to determine whether ^11^C-choline PET/CT imaging findings correlated with patient prognosis.

## 2. Results

### 2.1. Patients

Sixty consecutive patients resected for HCC have been considered for the study. Table 1 details the baseline characteristics. Of these, 14 (23%) patients were in stage 0-A, 37 (62%) in stage B, and 9 (15%) in stage C of the BCLC classification. Twenty-two (37%) patients had multiple tumors, and 37 (62%) had elevated alpha-fetoprotein before hepatectomy. Microvascular invasion and cirrhosis were histologically proven in 40 (67%) and 10 (22%) patients, respectively. However, the underlying liver function was globally normal as documented by the values of Bilirubin and Cholinesterases (BILCHE), Child–Pugh–Turcotte, and Model for End-stage Liver Disease MELD, scores [19,20,21,22]. Table 2 details the surgical data and short-term outcome. Most of the patients underwent minor hepatectomy, meaning IOUS-guided limited resection [23,24]. Overall postoperative complications were recorded in 24 (40%) patients, of which only 5 (21%) were graded as major complication (Clavien-Dindo 3–4) [25]. The 30-day mortality was nil.

### 2.2. Results of the PET/CT Parameters

Table 3 details the metabolic data subject of the study. The median values of the seven PET/CT parameters were as follows: SUVmax 15.2 (range 6.2–28.2), SUVmean 11 (SD ± 3), SUVliver 10.8 (range 5.6–16.6), MTV 33.4 (range 1.7–1665.8), metabolic tumor burden (MTB = MTV × SUVmean) 447.8 (range 13–12141), SUVratio 1.5 (range 0.63–3.12). Photopenic areas were present in 34 (57%) patients. Figure 1 details two representative cases of HCC staged using ^11^C-choline PET/CT.

### 2.3. Results of the Survival Analysis

After a median follow-up of 17.96 months (range 1.2–83.8), 12 patients (20%) died. The five-year OS was 61.5%. The median survival was not reached. During the follow-up, 17 (28.3%) developed HCC recurrence, which was treated with trans-arterial embolization. Table 4 details the results of the multivariate analysis of prognostic factors for OS. As reported, among MELD score, BILCHE score, BCLC stages, number of tumor, size of tumor, value of serum AFP, tumor grading, microvascular invasion, cirrhosis, SUVmax, SUVmean, SUVliver, MTV, photopenic areas, MTB and SUVratio, only the BCLC stages (HR = 2.94; 95%CI = 1.41–4.51; *p* = 0.003) and MTV (HR = 2.11; 95%CI = 1.51–3.45; *p* = 0.026) were found to be independently associated with OS. In particular, patients with larger MTV had two-fold risk of death during the follow-up.

### 2.4. Association between MTV and Survival

Following the previous findings, we performed receiver operating characteristics (ROC) curve analysis for MTV tested against the risk of death. With an area under the curve (AUC) = 0.77 (95%CI = 0.57–097; *p* < 0.001), we found that 380 mL was the best cutoff value (sensitivity 82%, specificity 62%) for the discrimination of overall survival after hepatectomy for HCC (Figure 2). Consistently, Figure 3 details the Kaplan–Meier curve for OS in relation to that cut-off value for MTV: the 3-year OS rates were 62% and 20% for patients with MTV < 380 and MTV ≥ 380, respectively (*p* = 0.015).

## 3. Discussion

The current guidelines for HCC management are heterogeneous. A recent review listed up to eight different guidelines, highlighting the lack of solid, high-level evidence [26]. Interestingly, most of these guidelines do not recommend PET/CT in the diagnostic work-up of HCC. At the same time, being the five-year survival reported between 40 and 75% after hepatectomy [3,4,5,6], it is clear that there is the need to refine the workup of HCC patients before surgery and in general before loco-regional treatments. Considering the tendency to operate even on patients with advanced HCC, such as those with macrovascular invasive and/or large or multinodular HCC, in dedicated centers [5,6,20], it is of paramount importance to precisely define the tumor burden before surgery. In addition, in these advanced HCC patients the risk of underestimation of the intra- and extrahepatic tumor burden is not negligible. In this sense, a diagnostic tool that allows a refinement of the indications for surgery in some specific advanced HCC patients might therefore be of outstanding value for clinicians involved in decision-making.

In the present study, we sought to investigate the eventual prognostic value of ^11^C-choline PET/CT in patients undergoing hepatectomy for HCC. In a cohort of 60 patients, who had PET/CT before surgery, only one nuclear medicine parameter was found to be independently associated with overall survival together with the BCLC stages. Of these two findings, the latter is not a novelty and not even a surprise. Indeed, the prognostic significance of the BCLC classification has been widely reported [3,4,5,6]. Conversely, the first finding that is the prognostic significance of the metabolic tumor volume represents an important novelty worthy of being spread among clinicians involved in the care of HCC patients. To the best of our knowledge, this is the first study reporting such an important role for ^11^C-choline PET/CT in stratifying the prognosis of HCC patients. Our work gives compelling evidence for measuring the MTV of ^11^C-choline PET/CT in any HCC patient awaiting loco-regional treatments to discriminate patients at risk of worse survival.

Our group previously reported two studies on the role of ^11^C-choline PET/CT in HCC patients. In the first report, we showed how this modality had good accuracy in investigating HCC patients in comparison to the CT and MRI, and how the main strength was its ability to detect extrahepatic localizations [16]. In the second study, as mentioned, we showed how the incorporation of ^11^C-choline PET/CT in the multidisciplinary discussion altered the decision-making process in up to one third of HCC patients [18]. Some other authors reported their experience with the more common FDG-PET/CT in HCC patients [27,28,29,30], showing how the value of such tracer could depend on the tumor grading. In moderate and well differentiated HCC, the FDG PET/CT may be negative because the tumor may not have increased glucose consumption in comparison with the adjacent non-tumoral liver tissue. In other words, a relatively high rate of false negative results may be expected using FDG, and similar experiences come from the use of different probes such as 18F-fluoroethylcholine, 18F-fluorocholine, and ^11^C-Acetate [31]. Thus, following our and other experiences, radiolabelled choline [11,12,13,14,15,32] emerged as a valid alternative to the standard FDG—especially for those patients with well-differentiated HCC.

In the light of the need to refine the selection criteria of HCC patient awaiting hepatic resection, the use of nuclear medicine may be a pivotal factor. Of note, only 23% of the patients included in this study were ideal candidate to hepatic resection according to the BCLC guidelines. The remaining 77%, that were in the BCLC stages B and C, should have been considered for other treatments modalities. While considering that the indication for surgery was decided in the multidisciplinary meeting as well as that the treatments output of the BCLC is somehow under debate, it is clear that tumor number and tumor size, which are the two main parameters that are considered when addressing a therapeutic output to a given patient with HCC, should be considered along other tumor features. Indeed, even though it might be a kind of interrelatedness between BCLC and MTV at least considering the collinearity of the tumor size, it might be the case that MTV reveals some more interesting metabolic features that cannot be revealed by considering only tumor size. Such metabolic features may reflect heterogeneity in tumor metabolic and/or genetic traits, and intratumoral metabolism that has been found to be driven by the oncogenic alteration involved in HCC tumorigenesis as recently reported by other authors [33]. In this sense, the values of MTV as measured by ^11^C-choline PET/CT may be of great help in identifying a subgroup of patients at risk of worse survival for which other treatments might be more indicated than surgical resection.

Similar original findings might emerge also by investigating the correlations between PET data and MRI data. This is certainly an emerging field of research since metabolism, perfusion and water diffusion may have a relationship in the same tumor, and their understanding could expand our knowledge of tumor characteristics [34].

This study has some limitations. First, this is a single-center retrospective study with relatively small sample size, thus, selection biases might be present. Second, the MTV cut-off value found in the present study should be validated in a larger and external patient group. Third, we did not consider the disease-free survival but only the overall survival, which is considered the most reliable, precise, and easy to be measured endpoint in cancer patients. However, further studies including also disease-free survival are recommended. Despite these limitations, this study is noteworthy in showing how ^11^C-choline PET/CT is important in terms of OS and may be beneficial in the workup of HCC patients awaiting loco-regional treatments.

## 4. Materials and Methods

### 4.1. Study Design and Data Collection

This is an observational retrospective study conducted on a prospectively maintained database of patients who underwent hepatectomy for HCC in a tertiary referral university hospital. The study protocol was in accordance with the ethical guidelines established in the 1975 Declaration of Helsinki. It was also in accordance with the procedures of the local ethical committee of the institution, and with the guidelines for the reporting of observational studies in epidemiology [35]. The study protocol was submitted to an international clinical trial registry (clinicaltrials.gov, registration number NCT03430635).

### 4.2. Study Endpoint

The study endpoint was the identification of those ^11^C-choline PET/CT parameters that might be predictors of OS.

### 4.3. Predictors of Survival

The following predictors of survival were considered for this study: the MELD score, which is the combination of serum bilirubin, serum creatinine, and the international normalized ratio (INR) for prothrombin time [19,20]; the Humanitas score, which is the combination of serum bilirubin and serum cholinesterases, liver stiffness as measured with FibroScan, presence of esophageal varices and type of hepatectomy [21,22]; the BCLC classification (by increasing in stage) (3); the number of tumors (single versus multiple); the size of tumor (by increasing of 1 cm); the serum alpha-fetoprotein (normal value versus elevated); the grading of tumor (G1–2 versus G3–4); the microvascular invasion (present versus absent); the cirrhosis (present versus absent). For the purpose of the study we also considered the following ^11^C-choline PET/CT metabolic data: maximum standardized uptake value (SUVmax), mean standardized uptake value (SUVmean), liver standardized uptake value (SUVliver), SUVratio (SUVmax/SUVliver), metabolic tumor volume (MTV), metabolic tumor burden (MTB), computed as a product of MTVxSUVmean [36,37], and presence ofphotopenic area on ^11^C-choline PET/CT.

### 4.4. Study Selection Criteria

The institutional prospectively maintained liver surgery unit database was queried for patients with HCC preoperatively staged with ^11^C-choline PET/CT between 2012 and 2018. This metabolic imaging modality was performed in addition to chest CT, and to abdominal CT or liver-specific magnetic resonance imaging (MRI). Inclusion criteria were the following: first diagnosis and first treatment of HCC; performance of hepatectomy without microscopic residual of tumor (R0); minimum follow-up of 12 months; complete clinical, surgical, pathological, and follow-up data. The exclusion criteria were the following: recurrent HCC; non-radical hepatectomy (R1 or R2); extrahepatic disease; patients without PET/CT; patients with missing data.

### 4.5. Preoperative Workup and Selection Criteria for Hepatectomy

The preoperative workup consisted of CT and/or liver-specific MRI, and^11^C-choline PET/CT, which were performed within 30 days of surgery in each patient. The therapeutic management of each patient was collegially discussed in multidisciplinary meeting with surgeons, hepatologists, oncologists, radiologists, nuclear medicine specialists, and radiotherapists. The patients were selected for hepatectomy based on consolidated published criteria [23,38]. Briefly, the absolute resection contraindications for HCC included encephalopathy, ascites, and serum bilirubin level greater than 2 mg/dL. Concomitant esophageal varices were not considered a contraindication once endoscopic treatment were performed. Among the patients with serum bilirubin level below 2 mg/dL, the value for remnant liver volume was set at 50%. In the event of insufficient residual volume, portal vein embolization was considered [39] Parenchymal-sparing techniques were systematically adopted to minimize liver sacrifice [24]. Patients potentially candidable to liver transplantation were sent for consultation in a local liver transplant center.

### 4.6. ^11^C-choline PET/CT

The radiopharmaceutical was synthetized using a General Electric TracerLab FXc module (General Electric Healthcare, Waukesha, WI, USA). Patients in a fasting state of at least 4 h received a total amount of 250–450 MBq of 11C-choline. Ten minutes later, whole-body axial images were obtained with an integrated PET/CT tomograph, either a Siemens Biograph LS 6 scanner (Siemens Medical Systems, Erlangen, Germany) or a GE Discovery PET/CT 690 (General Electric Healthcare). Images were reconstructed and acquired according to the European Association of Nuclear Medicine Research Ltd. standardization program (http://www.eanm.org) to minimize differences in semi-quantitative evaluations related to the use of two different scanners. All scan volumes were obtained from the skull base to the mid-thigh and reconstructed in axial, sagittal, and coronal planes. Imaging data were reviewed by an experienced nuclear medicine physician. Seven PET/CT parameters were recorded: maximum standardized uptake value (SUVmax), mean standardized uptake value (SUVmean), liver standardized uptake value (SUVliver), metabolic tumor volume (MTV), photopenic area, metabolic tumor burden (MTB = MTV × SUVmean), and SUVratio (SUVmax/SUVliver). Regional lymph nodes were considered pathological when they measured over 10 mm, had a rounded shape, and showed increased tracer uptake. Furthermore, any unprecedented increased uptake in other organs, in the bone or the bone marrow, was considered malignant unless proven otherwise.

### 4.7. Follow-Up

The follow-up consisted of physical examination, blood tests with alpha fetoprotein testing, ultrasonography, CT scan, or MRI every 3 months after surgery.

### 4.8. Statistical Analysis

Categorical variables were reported as numbers and percentages. Continuous variables were reported as mean and standard deviation (SD). Because normal distribution could not be confirmed for any variable, nonparametric statistical tests were preferentially used. Survival data were obtained with the Kaplan–Meier method and compared using the log-rank test. The Cox-proportional hazard model was used to identify independent prognostic factors, including PET/CT parameters, for OS that was defined as the difference between the date of hepatectomy and the date of the last follow-up available or death. receiver operating characteristics (ROC) curve analysis was employed to assess the discrimination ability of variable(s) with respect to the occurrence of events for OS. Youden’s *J* statistic was used to determine the optimal cut-off value with the corresponding area under the curve (AUC), *p*-value, sensitivity, and specificity. Results were considered significant if *p* < 0.05. Analyses were computed by using software PRISM and IBM-SPSS.

## 5. Conclusions

This study demonstrated that adding ^11^C-choline PET/CT to the preoperative workup of HCC patients awaiting hepatic resection may lead to identify those patients at risk of worse survival. Among several different parameters, only MTV seemed to be a valuable prognostic parameter derived from ^11^C-choline PET/CT. In addition, ^11^C-choline PET/CT added to the prognostic value of the BCLC classification strengthening its prognostic ability. Incorporation of such metabolic modality into HCC stating system may be beneficial and should be considered.

## Figures and Tables

**Figure 1 cancers-13-00472-f001:**
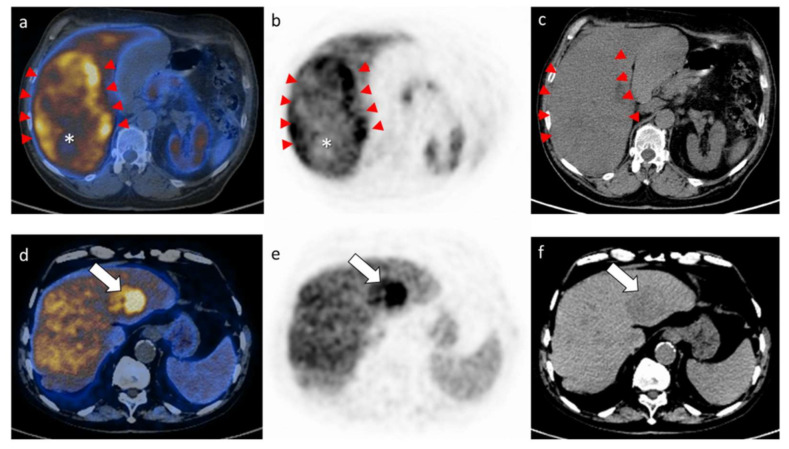
Illustration of two explicatory examples presenting with diverse patterns on PET images, herein shown with fused PET/CT (**a**,**d**), PET alone axial views (**b**,**e**) and low dose CT (**c**,**f**) at the level of the HCC lesion. The upper panels (**a**–**c**) represent a patient with a large HCC characterized by heterogeneous Choline uptake, mostly located in the tumor borders (red arrowheads) and associated to a clear photopenic area on the fused PET/CT and PET alone axial views (**a**,**b**; while asterisk). The SUVmax (maximal standardized uptake value) of the lesion resulted 25.4. The lower panels (**d**–**f**) represent a patient with a smaller HCC lesion characterized by an overall homogeneous and intense choline uptake, as indicated by the white arrows on all axial views. The SUVmax of the lesion herein resulted 20.2.

**Figure 2 cancers-13-00472-f002:**
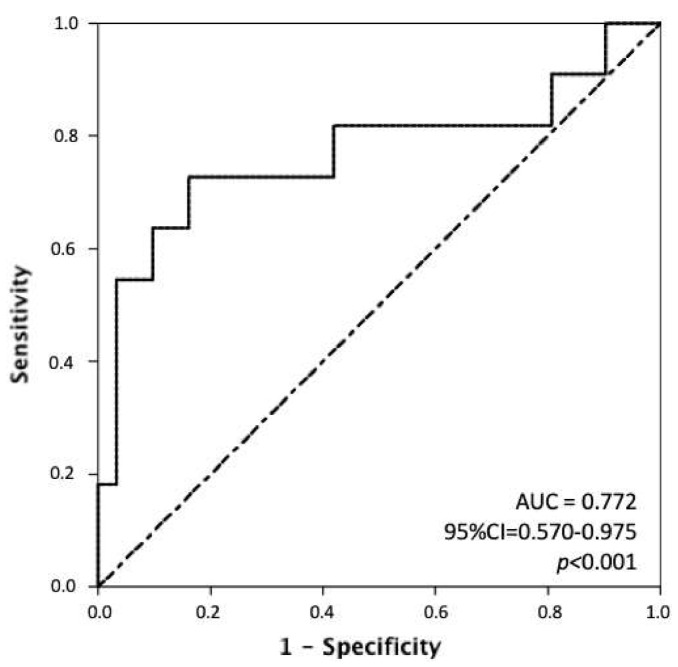
The figure shows the Receiver Operating Characteristics (ROC) curve analysis of metabolic tumor volume (MTV) for the prediction of overall survival. With an area under the curve (AUC) = 0.77 (95%CI = 0.57–0.97; *p* < 0.001), we found that 380 mL was the best cutoff value (sensitivity = 82%, specificity = 62%) for the discrimination of overall survival after hepatectomy for HCC.

**Figure 3 cancers-13-00472-f003:**
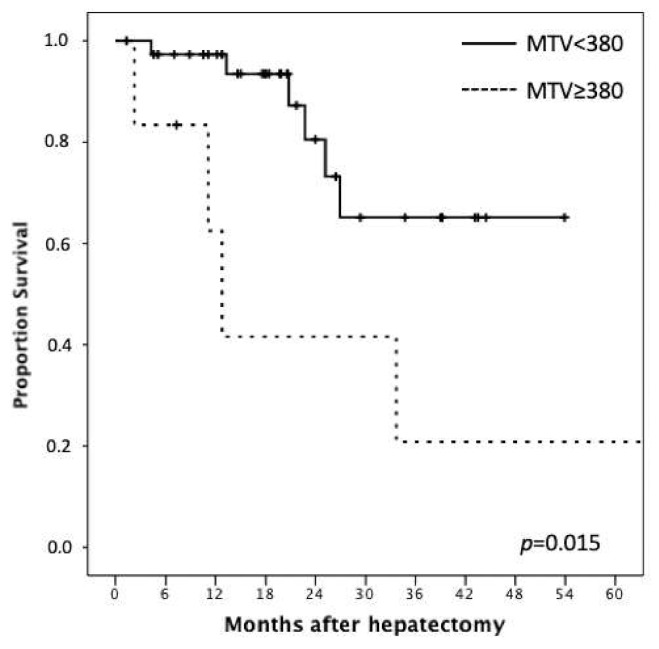
The figure shows the Kaplan–Meier curve for OS in relation to cut-off value for metabolic tumor volume (MTV): the 3-year OS rates were 62% and 20% for patients with MTV < 380 and MTV ≥ 380, respectively (*p* = 0.015).

**Table 1 cancers-13-00472-t001:** Baseline characteristics.

Characteristic	*No*. (%)
Age (years)	
Median, range	72 (18–83)
Gender	
Men	47 (78%)
Women	13 (22%)
BILCHE score	
0	35 (58%)
1	8 (13%)
2	10 (17%)
3	7 (12%)
CPT score A	60 (100%)
MELD score	
Median, range	8 (6–11)
BCLC stage	
0–A	14 (23%)
B	37 (62%)
C	9 (15%)
Tumor number	
Median, range	1 (1–8)
>1	22 (37%)
1	38 (63%)
2	17
3	1
4	2
6	1
8	1
Tumor size (cm)	
Median, range	7.5 (1.4–26)
Milan criteria	
In	12 (20%)
Out	48 (80%)
Alpha-fetoprotein (ng/mL)	
Median, range	17 (2–45,667)
>7 *	37 (62%)
Tumor grading	
1–2	34 (57%)
3–4	26 (43%)
Microvascular invasion	
No	20 (33%)
Yes	40 (67%)
Cirrhosis	
No	47 (78%)
Yes	10 (17%)
Unknown	3 (5%)

* Upper limit of the range in our laboratory. Abbreviations: BILCHE, Bilirubin and Cholinesterases score; CPT, Child-Pugh-Turcotte; MELD, Model for End-stage Liver Disease.

**Table 2 cancers-13-00472-t002:** Surgical and short-term outcome data.

Data	*No*. (%)
Type of hepatectomy	
Minor resection	51 (85%)
Major resection (>3 segments)	9 (15%)
Thoracoabdominal approach	22 (37%)
Blood loss (mL)	
Median, range	350 (20–3000)
Blood transfusions	11 (18%)
Complications	
Overall	24 (40%)
Minor (Dindo-Clavien 1–2)	19 (79%)
Major (Dindo-Clavien 3–4)	5 (21%)
Mortality	
30-day	0

**Table 3 cancers-13-00472-t003:** Metabolic ^11^C-choline PET/CT data.

Parameter	Value
SUV max	
Median; range	15.2 (6.2–28.2)
Mean; SD	16.4 (±5.6)
SUV mean	
Mean; SD	11 (±3)
SUV liver	
Median; range	10.8 (5.6–16.6)
Mean; SD	10.9 (±2.4)
MTV	
Median; range	33.4 (1.7–1665.8)
Mean; SD	185.9 (±375.9)
Photopenic areas (yes)	34 (57%)
MTB	
Median; range	447.8 (13–12,141)
Mean; SD	1593.9 (±2697.6)
SUV ratio	
Median; range	1.5 (0.63–3.12)
Mean; SD	1.5 (±0.6)

**Table 4 cancers-13-00472-t004:** Multivariate analysis of prognostic factors for overall survival (OS).

Factor	HR	95% CI	*p*-Value
MELD score	1.03	0.58–2.12	0.138
BILCHE score	1.22	0.91–2.91	0.235
BCLC classification	2.94	1.41–4.51	0.003
Number of tumor	1.26	0.81–1.96	0.087
Size of tumor	1.18	0.91–1.99	0.094
AFP	0.91	0.49–1.91	0.814
Tumor grading	1.12	0.81–1.71	0.331
Microvascular invasion	1.33	0.32–1.51	0.393
Cirrhosis	0.81	0.39–1.81	0.941
SUVmax	0.86	0.39–1.61	0.569
SUVmean	0.87	0.70–1.07	0.190
SUVliver	0.84	0.41–1.12	0.640
MTV	2.11	1.51–3.45	0.026
Photopenic areas	0.52	0.06–1.39	0.527
MTB	0.99	0.89–1.79	0.213
SUVratio	1.18	0.91–2.96	0.336

## Data Availability

The data presented in this study are available on request from the corresponding author.

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
