# Peer review of "Prognostic Value of Metabolic Imaging Data of 11C-choline PET/CT in Patients Undergoing Hepatectomy for Hepatocellular Carcinoma"

_cancers, 2021, doi:10.3390/cancers13030472_

Round 1
Reviewer 1 Report
Thank you for the revision, the manuscript is ready to be published in its current form.
Reviewer 2 Report
Interesting paper and hopefully applicable widely soon. In the mean time, it would be great if the results could be confirmed on more patients or at more sights. Looking forward to reading about it more.
Author Response
We thank the reviewer for her/his notes. We agree that our findings may be soon applied in other centers, and hopefully in that way we could have more data on ¹C-choline PET/CT data in HCC patients.
Reviewer 3 Report
Title:
Prognostic value of metabolic imaging data of 11C-choline PET/CT in patients undergoing hepatectomy for hepatocellular carcinoma
Summary:
This is a well done, well-written study retrospectively evaluating whether 11C-Choline PET/CT is associated with OS in patients with HCC who underwent subsequent resection. This study found the BCLC staging and metabolic tumor burden are associated with OS. Given the lackluster performance of FDG PET in this disease, new metabolic imaging agents are needed and authors help to reaffirm existing work with 11C-Choline in patients with HCC.
Overall, this work would be a compelling addition to literature.
This reviewer has minor edits/revisions/suggestions:
Minor
Simple summary:
Page 1, line 16: Recommend edit: “The aim *of this study* was to…”
Abstract:
Page 1, Line 17: font for “metabolic imaging” is different than rest of text
Minor word choice/phrasing edits would be helpful for clarity (e.g. page 1, line 26 could be edited to something like “We queried our prospective database for patients with HCC staged with 11C-choline PET/CT to assess the clinical impact of this imaging modality.”
Results
Authors may want to include any subsequent treatments (systemic and/or local) which may affect survival (TKI or atezo/bev) in their cohort to confirm the groups are balanced in that dimension.
For FDG PET, MTV x SUVmean is designated as “total lesion glycolysis.” Authors should consider introducing this parallel idea as related to the metabolic burden parameter used here to reinforce that this idea has been used elsewhere and support the concept here.
Page 2, line 62, wording suggestion: “In this study, we sought to determine whether 11C-choline PET/CT imaging findings correlated with patient prognosis”
Figure 1:
- Authors should describe the imaging modality used in each panel (e.g. A: PET/MR, B: PET/CT).
- B: arrows are displaced in left and right panels
- Authors use different color gradients for PET in panel a, and b. Figure could benefit from more cohesive organization. For example, If PET/MR and PET/CT have been selected. Authors could choose to present an MR sequence that shows the lesion in panel A, left, and the MR/PET merge on panel B, right, as they have done in panel B.
- Authors can consider similar magnification of figures/panels
- Are there any clinical features of lesion in patient in panel A and B that might explain the differences seen in SUV?
- Are the SUVs in figure SUVmean or SUVmax?
Given that BCLC classification status and MTV showed significant association with OS, it would be helpful to see how well BCLC classification correlated in this cohort and how this compared on ROC, and placed along side figure 2. Same comment regarding the KM in Figure 3.
Page 7, line 182: Given that this is a retrospective study, reviewer suggests softening this language on whether 11Choline PET/CT “should” be in added to “may be beneficial”
Discussion:
Authors should comment on the interrelatedness of MTV and BCLC classification, especially since BCLC includes a nodule size component which is likely to correlated with MTV.
Discussion may benefit from inclusion of other PET imaging agents that are being investigated in HCC for similar ends, including 11C-acetate, 18F-FLT, 18-Flourocholine, 18FDGal) and perhaps PSMA imaging agents. In addition, several groups have looked at combining 2 of these agents to improved the sensitivity/specificity for HCC… to the best of this reviewer’s knowledge, authors of this manuscript are the first to try to correlated the new imaging agents to OS.
Methods:
Please include the range of years during which the patients included in this analysis were treated.
Page 7, line 192, wording: suggest deleting “Besides”
Conclusions:
Given that this study is not prospective, authors should soften conclusions on prognostic value of choline PET/CT. However, that the choline PET/CT parameter of MTV correlates to OS as does BCLC staging is very compelling and should be explicitly underscored here.
Author Response
Overall, this work would be a compelling addition to literature.
Authors reply: We thank the reviewer for her/his suggestions. Please find the new revised manuscript as well as our replies here.
Page 1, line 16: Recommend edit: “The aim *of this study* was to…”
Authors reply: We thank for this note. We have now modified accordingly.
Abstract:
Page 1, Line 17: font for “metabolic imaging” is different than rest of text
Authors reply: We do not know why this happened. However, we have now changed accordingly.
Minor word choice/phrasing edits would be helpful for clarity (e.g. page 1, line 26 could be edited to something like “We queried our prospective database for patients with HCC staged with 11C-choline PET/CT to assess the clinical impact of this imaging modality.”
Authors reply: We thank for this suggestion, which has been taken. Now the sentence has been accordingly modified.
Results
Authors may want to include any subsequent treatments (systemic and/or local) which may affect survival (TKI or atezo/bev) in their cohort to confirm the groups are balanced in that dimension.
Authors reply: This is an important note. Please be advised that none of these 60 patients had postoperative systemic therapy. Some of them, 17 (28%) underwent transarterial embolization because of multifocal recurrence. This datum has been now added in the manuscript.
For FDG PET, MTV x SUVmean is designated as “total lesion glycolysis.” Authors should consider introducing this parallel idea as related to the metabolic burden parameter used here to reinforce that this idea has been used elsewhere and support the concept here.
Authors reply: We thank for this note, and we completely agree with the reviewer. The concept of TLG perfectly fits the case and has inspired the use of metabolic burden for HCC investigated with Choline PET. We have now introduced MTB (metabolic tumor burden) as equivalent for total lesion activity (TLA) or TLG in Choline PET (MTV x SUVmean). The references have been adapted accordingly. Thanks.
Page 2, line 62, wording suggestion: “In this study, we sought to determine whether 11C-choline PET/CT imaging findings correlated with patient prognosis”
Authors reply: Thank your for this suggestion, following which we have now modified the text.
Figure 1:
- Authors should describe the imaging modality used in each panel (e.g. A: PET/MR, B: PET/CT).
- B: arrows are displaced in left and right panels
- Authors use different color gradients for PET in panel a, and b. Figure could benefit from more cohesive organization. For example, If PET/MR and PET/CT have been selected. Authors could choose to present an MR sequence that shows the lesion in panel A, left, and the MR/PET merge on panel B, right, as they have done in panel B.
- Authors can consider similar magnification of figures/panels
- Are there any clinical features of lesion in patient in panel A and B that might explain the differences seen in SUV?
- Are the SUVs in figure SUVmean or SUVmax?
Authors reply: Based on the suggestions of the reviewer, we have modified the figure by using the same color gradient, the same pannel distribution, the some modality (PET/CT) for fused images, and the same magnification. Arrows and other highlights have been adapted accordingly. The figure description has been changed to better illustrate the images. All SUVs used refer to SUVmax values.
Given that BCLC classification status and MTV showed significant association with OS, it would be helpful to see how well BCLC classification correlated in this cohort and how this compared on ROC, and placed along side figure 2. Same comment regarding the KM in Figure 3.
Authors reply: This is interesting. However, please be advised that the aim of the paper was to investigate the prognostic role of metabolic tumor features by using 11Choline PET/CT. We did not want to make comparative analysis with staging system such as the BCLC classification, which was anyway considered as one of the covariates to give to the reader the information on the tumor burden of the included patients as a descriptive feature. The fact that this classification came significant is not unexpected, but at the same time we do not have the statistical power here to place side by side the statistical, clinical and prognostic weight of that classification versus the weight of the MTV.
In regard to the KM, please be advised that the BCLC classification includes four stages, of which three are here present (stage 0-A, B and C). Their representations as three different curves would be difficult to be accepted being the total sample of patients 60. Moreover, as said just above, the comparison between BCLC and MTV in their ability in predicting HCC patients prognosis was not one of our endpoints. This might be considered in the next future. Thank you for this suggestions.
Page 7, line 182: Given that this is a retrospective study, reviewer suggests softening this language on whether 11Choline PET/CT “should” be in added to “may be beneficial”
Authors reply: We thank for this suggestion. We have accordingly modified that sentence.
Discussion:
Authors should comment on the interrelatedness of MTV and BCLC classification, especially since BCLC includes a nodule size component which is likely to correlated with MTV.
Authors reply: Following this note, we have now added a paragraph in the Discussion. While there might be a interrelatedness between MTV and BCLC classification, exactly based on the tumor size component, please consider that the metabolic activity expressed with MTV by using C-choline PET/CT should be further investigated with metabolomics data, which unfortunately we do not have here. We make treasure of this note for future studies.
Discussion may benefit from inclusion of other PET imaging agents that are being investigated in HCC for similar ends, including 11C-acetate, 18F-FLT, 18-Flourocholine, 18FDGal) and perhaps PSMA imaging agents. In addition, several groups have looked at combining 2 of these agents to improved the sensitivity/specificity for HCC… to the best of this reviewer’s knowledge, authors of this manuscript are the first to try to correlated the new imaging agents to OS.
Authors reply: We thank for this note, and please be advised that we have now modified our discussion by including these suggestions.
Methods:
Please include the range of years during which the patients included in this analysis were treated.
Authors reply: We have now added the study period, which was from 2012 to 2018.
Page 7, line 192, wording: suggest deleting “Besides”
Authors reply: We have now modified accordingly.
Conclusions:
Given that this study is not prospective, authors should soften conclusions on prognostic value of choline PET/CT. However, that the choline PET/CT parameter of MTV correlates to OS as does BCLC staging is very compelling and should be explicitly underscored here.
Authors reply: We thank for this note. We have soften the conclusions and overall the manuscript. Thank for the opportunity to revise the paper.
Round 2
Reviewer 1 Report
Thank you for the opportunity to review this interesting paper.
The authors sought to elucidate the prognostic value of Choline-PET in HCC patients undergoing surgery.
Only the MTV parameters derived from PET and BCLC stages remained statistically significant for prediction of overall survival. This is especially of interest as most commonly used SUV values did not remain significant.
As stated by the authors, PET-CT is not yet recommended in the guidelines, as especially FDG-PET CT suffers from the metabolic activity of the liver itself.
I have only minor concerns for this study:
-did the patients reveice any form of treatment before the surgery (TACE, RFA)? This could have an influence on the prognosis.
-You mentioned that the number of tumors was divided to single versus multiple. Did all these patients were within the Milan criteria? So you should divide the patients according to the exact tumor number.
-How was tumor size exactly measured? There might be differences, whether the multi-phase CT or MRI was used. You could also calculate the axial size of the PET.
-In your figure 1, the first image is a fused PET/MRI image (of the heaptobiliary phase). As you did not use a PET/MRI please use a PET/CT image. Did you perform a fusion with the MRI images in all patients? Were there better spatial correlations with the PET image?
Moreover, the arrows for the PET/CT images are not showing the HCC lesions. In your PET/CT image, you used a non-enhanced low dose CT image. Did you perform a diagnostic CT simultaneosly to the PET?
Please use a more precise statement in your conclusion that only MTV seems to be a valuable parameter derived from Choline PET-CT in HCC patients.
Could you correlate your PET findings with MRI data? Such as Diffusion-weighted imaging or the HPB sequence? Presumably, the best of both worlds would better predict prognosis than only the PET information itself.
-
Author Response
We thank the reviewer her/his notes. Please find the new manuscript with changes and revisions made accordingly.
-did the patients reveice any form of treatment before the surgery (TACE, RFA)? This could have an influence on the prognosis.
Authors reply: We thank for this note. None of the included patients had previous treatment for HCC. All these patients were at the first diagnosis and first treatment. This has been now clarified in the inclusion/exclusion study criteria.
-You mentioned that the number of tumors was divided to single versus multiple. Did all these patients were within the Milan criteria? So you should divide the patients according to the exact tumor number.
Authors reply: Following this note, we have now modified the text of the manuscript and the date in table 1. Among 60 patients, 38 had single tumors and 22 had more than 1 tumor. Among these 22 patients with multiple tumors, 17 had two tumors, 1 had 3 tumors, 2 had 4 tumors, 1 had six tumors and finally 1 had 8 tumors. We have also classified patients according with the Milan Criteria. Please find these data in Table 1. Yet, please be advised that none of the patients here reported were transplantable upfront. The age of the patients was 72 years old, and those younger patients were not considered for transplantation because of advanced disease or because they refuted such operation. As said in the manuscript, the indication for hepatectomy, rather than any other treatment, was concorded during our MDT meeting at the presence among other hepatologists and medical oncologists.
-How was tumor size exactly measured? There might be differences, whether the multi-phase CT or MRI was used. You could also calculate the axial size of the PET.
Authors reply: We thank for this note. Please be advised that the tumor size was calculated on the specimen by the pathologist. We considered the tumor size, as well as the tumor number for instance, that recorded on the specimen.
-In your figure 1, the first image is a fused PET/MRI image (of the heaptobiliary phase). As you did not use a PET/MRI please use a PET/CT image. Did you perform a fusion with the MRI images in all patients? Were there better spatial correlations with the PET image?
Authors reply: We thank for this observation. As required, we have now changed the figure 1 by using PET/CT images for both examples. The arrows and other highlights have been adapted accordingly. With regards to fusion with MRI, it was not performed routinely. We did not use either diagnostic or ceCT simultaneously with PET. MRI, as well as ceCT, were used as comparison to ameliorate in some cases HCC delineation.
Moreover, the arrows for the PET/CT images are not showing the HCC lesions. In your PET/CT image, you used a non-enhanced low dose CT image. Did you perform a diagnostic CT simultaneosly to the PET?
Authors reply: Yes, please find the new Figure 1 that has been accordingly changed. With regard to the diagnostic CT, we did not perform these two imaging modalities simultaneously.
Please use a more precise statement in your conclusion that only MTV seems to be a valuable parameter derived from Choline PET-CT in HCC patients.
Authors reply: We have now modified our conclusions both in the abstract and in the main text.
Could you correlate your PET findings with MRI data? Such as Diffusion-weighted imaging or the HPB sequence? Presumably, the best of both worlds would better predict prognosis than only the PET information itself.
Authors reply: We thank the reviewer for this very important question. Unfortunately, we cannot now correlate PET data with MRI data because this would require a dedicated study. Please be advised that we can plan this study soon, and in this sense we have now added a sentence as a comment in the discussion.